

# Sperm specificity and potential paternal effects in gynogenesis in the Amazon Molly (*Poecilia formosa*)

Clarissa Cerepaka[1,2,3] and Ingo Schlupp[2,4]

[1] Division of Comparative Medicine, University of Oklahoma Health Sciences Center, Oklahoma City, Oklahoma, United States of America
[2] Department of Biology, University of Oklahoma, Norman, Oklahoma, United States of America
[3] Lab Animal Resource Center, The University of Texas at Dallas, Richardson, TX, United States of America
[4] International Stock Center for Livebearing Fishes, University of Oklahoma, Norman, Oklahoma, United States of America

Corresponding author
Clarissa Cerepaka,
clarissa.cerepaka@utdallas.edu

## ABSTRACT

The Amazon Molly (*Poecilia formosa*) reproduces by gynogenesis, a relatively rare form of asexual reproduction where sperm is required to trigger embryogenesis, but male genes are not incorporated into the genome of the embryo. Studying gynogenesis could isolate paternal non-genetic effects on reproduction. This study explored which of eleven related species can produce sperm to trigger gynogenesis through natural mating in *P. formosa*, and whether sympatry affects reproductive success in *P. formosa*. Reproductive outcomes measured were relative reproductive output (number of offspring in the first brood divided by female standard length), relative embryo output (number of embryos in the first brood divided by female standard length) and combined relative reproductive output (sum of relative reproductive output and relative embryo output). For large (>4 cm) *P. formosa*, combined relative reproductive output was higher with sympatric Atlantic Molly (*Poecilia mexicana*) males than with allopatric *P. mexicana* males. *P. formosa* produced live offspring or late-stage embryos with all species tested in the genera *Poecilia* and *Limia* but did not produce offspring or embryos with males from the genera *Gambusia*, *Girardinus*, *Heterandria*, *Poeciliopsis*, or *Xiphophorus*. This information, as well as the limitations characterized in this study, will set a foundation for use of *P. formosa* as a model for paternal effects and the species specificity of sperm on fertilization, embryogenesis, and reproductive success.

# INTRODUCTION

Gynogenesis is a relatively rare form of asexual reproduction where sperm is required to trigger embryogenesis, but male genes are not incorporated into the genome of the embryo (*Schlupp, 2010*; *Lehtonen et al., 2013*). This form of reproduction is known from several invertebrates (*Suomalainen, Saura & Lokki, 1987*) and a small number of vertebrates

(*Avise, 2008*), where it is always associated with a hybrid origin. There are two more forms of reproduction where females rely on sperm of heterospecific males to trigger embryogenesis, kleptogenesis (*Bogart, 2019*) and hybridogenesis (*Vrijenhoek, 1979*). What is puzzling in gynogenesis is that egg and sperm *do* interact and fuse, as evidenced by occasional introgression of male genes (*Nanda et al., 1995*; *Schartl et al., 1995*; *Warren et al., 2018*), but mostly sperm DNA is not incorporated, and the sperm is only serving to trigger embryogenesis.

Species that reproduce by gynogenesis provide an unparalleled opportunity to study if sperm affects reproductive success without the incorporation of male genes. Sperm dependency in gynogenesis raises several questions at different levels of analysis. First, we have a very limited understanding of the physiological and reproductive biology aspects of sperm dependency, although this is not only crucial for our understanding of gynogenesis, but also touches on sperm-egg interactions in general, including infertility in humans. Second, we also have a poor understanding of the evolutionary and phylogenetic dimension of sperm dependency. In some gynogenetic species, sperm from multiple species are capable of triggering embryogenesis (*Bogart, 2019*). What allows some sperm to trigger embryogenesis while other sperm does not seem to work? Is phylogenetic distance or divergence time a factor in this? Finally, does the presence or absence of sperm-providing (donor) species influence where gynogenetic species can live? The present study explored these questions.

Although generally uncommon, there are several independent origins of gynogenesis within the teleost fishes. Fish genera that include hybrid species which reproduce by gynogenesis include *Poecilia, Poeciliopsis, Chrosomus (formerly Phoxinus), Misgurnus, Fundulus, Menidia, Squalius, Cobitis*, and *Carassius* (*Vrijenhoek, 1979*; *Dawley, 1992*; *Beukeboom & Vrijenhoek, 1998*; *Schlupp, 2005*; *Lafond et al., 2019*; *Dalziel et al., 2020*; *Vogt, 2021*). In *Poeciliopsis* and *Chrosomus*, species also reproduce by hybridogenesis, which is also characterized by sperm dependency, while showing a different mechanism of gamete formation (*Vrijenhoek, 1979*; *Lafond et al., 2019*; *Vogt, 2021*). *Misgurnus, Cobitis, Squalius*, and *Carassius* include species which naturally reproduce by gynogenesis, but these same species frequently incorporate sperm genomes into some of the offspring, creating polyploid offspring in reproductive processes other than gynogenesis (*Itono et al., 2007*; *Janko et al., 2007*). *Fundulus* hybrids between *F. heteroclitus* and *F. diaphanus* and the species *Menidia clarkhubbsi* appear to reproduce almost exclusively by gynogenesis, and fertilization occurs externally (*Echelle, Echelle & Crozier, 1983*; *Dawley, 1992*; *Dalziel et al., 2020*). Within amphibians, the genus *Ambystoma* utilizes gynogenesis as one of several mechanisms that contribute to asexual reproduction by kleptogenesis. In *Ambystoma*, for example, sperm of five different species can trigger embryogenesis, but to our knowledge an experiment as reported here has not been conducted in any other asexual species (*Bogart, 2019*).

Among fish species reproducing by gynogenesis, the all-female, viviparous Amazon Molly (*Poecilia formosa*) is unique in combining a diploid genome, live-bearing, and strict gynogenesis. In addition to viviparity and gynogenesis, *P. formosa* is also a hybrid species between a *Poecilia mexicana* female and a *Poecilia latipinna* male that has continued to

thrive for 120,000–280,000 years since the single hybridization event (*Turner, Brett & Miller, 1980*; *Schlupp, 2005*; *Schlupp, 2010*; *Warren et al., 2018*). Studies of paternal introgression in *P. formosa* have concluded that introduction of small amounts of paternal DNA during reproduction was unlikely to have contributed significantly to natural selection or evolution of *P. formosa* (*Warren et al., 2018*; *Dedukh et al., 2022*). In the absence of significant male genetic contribution, *Warren et al. (2018)* suggested that *P. formosa* may persist because of its specific genomic combination conferring ideal evolutionary fitness, or intraspecific competition may select for clones with advantageous mutations. Whether paternal non-genetic effects could affect reproductive success, natural selection, or evolution in *P. formosa* has not been studied yet. The evolutionary advantage of strict gynogenesis remains unclear, as gynogenetic species are ecologically limited by their sperm donor host species (*Schlupp, 2005*). In another gynogenetic complex, asexual reproduction may be a strategy to overcome hybrid sterility as demonstrated by gynogenetic *Cobitis elongatoides* and *C. taenia* hybrids (*Dedukh et al., 2020*). As genetic divergence in the parental species increases, the karyotype of the hybrid gynogenetic species has greater heterozygosity providing greater hybrid vigor (*Marta et al., 2023*). However, viable hybrid species, such as *P. formosa*, suggest that interspecies reproduction may not always be highly specific. Hybridization has to overcome both prezygotic and postzygotic reproductive barriers, and these constraints often limit future reproduction of the hybrid species (*Coughlan & Matute, 2020*). *P. formosa* escapes most intrinsic postzygotic barriers by preserving high heterozygosity through gynogenesis. However, the reproductive success of *P. formosa* may still be constrained by extrinsic and intrinsic prezygotic barriers such as male mate choice, availability of sperm donor species as hosts, physical and chemical signaling during copulation, and sperm-egg compatibility (*Schlupp, 2005*; *Schlupp & Plath, 2005*; *Gabor & Grober, 2010*). In a natural setting, *P. mexicana* and *P. latipinna* are the primary sperm donors to *P. formosa*, but both species preferentially mate with conspecifics rather than *P. formosa*. More rarely, *P. latipunctata* and *P. sphenops* are additional sperm donors to *P. formosa* observed in nature (*Schlupp, 2009*). Other species from the genera *Poecilia* and *Limia*, including *P. velifera, P. petenensis, P. sulphuraria, P. reticulata*, and *Limia vittata* are all reported sperm donors from laboratory crosses in published literature (*Schultz & Kallman, 1968*; *Schlupp, Parzefall & Schartl, 2001*; *Schlupp, 2010*).

Fertilization in *P. formosa* has not been directly observed, but the spermatozoon does appear to have a key role. Fertilization and gestation are intrafollicular within the ovary and sperm can be observed in the folds of the ovarian lumen throughout all stages of gestation and nongestation (*Uribe et al., 2016*). Meiosis in the oocyte arrests during prophase I preventing synapsis of homologous chromosomes, consistent with apomixis (suppression of meiosis) until signaled, presumably by sperm interaction, to divide mitotically (*Dedukh et al., 2022*). In another gynogenetic fish, *Carassius gibelio*, a cyprinid widespread in Asia and Europe (*Freyhof & Kottelat, 2007*), males of several congeneric species have been described to serve as natural sperm donors (*Fan & Shen, 1990*) including other members of the speciose family Cyprinidae such as the more distantly related genera *Rutilus* (*Paschos et al., 2004*), and *Megalobrama* (*Yi et al., 2003*). In artificially induced

gynogenesis in aquaculture species, ultraviolet-irradiated sperm with damaged genetic material successfully induced gynogenesis in species from disparate genera (Nile tilapia *Oreochromic niloticus* and jaguar cichlid *Parachromis managuensis*, red crucian carp *Carassius auratus* and common carp *Cyprinus carpio* or blunt snout bream *Megalobrama amblycephala*, rainbow trout *Oncorhynchus mykiss* and sea trout *Salmo trutta* (*Liu, 2010*; *Polonis et al., 2018*; *Cao et al., 2021*). Gynogenesis induced by ultraviolet-irradiated sperm suggests that sperm-egg interaction is less specific in gynogenesis than sexual reproduction because the sperm genome is not incorporated into the embryo. In zebrafish, early embryonic development can be induced by UV-irradiated *C. carpio* sperm, but the embryo remains haploid (*Delomas & Dabrowski, 2016*). These examples support that sperm-egg interactions in gynogenesis may not be highly specific.

Several clonal lineages of *P. formosa* exist in different geographic locations ranging from central Mexico, the mouth of the Rio Tuxpan where the initial hybridization event occurred near Tampico, Mexico, to the northern limit of the Nueces River in Texas (*Schlupp, Parzefall & Schartl, 2001*; *Schlupp, 2010*), and more recently to the Brazos River near Houston, Texas (*Martin, Cohen & Hendrickson, 2012*). While both parental species, *P. mexicana* and *P. latipinna*, exist in Mexico, the Texas *P. formosa* populations breed with *P. latipinna* males. *P. latipinna*, the paternal ancestor, ranges from North Carolina to the Rio Tuxpan, and *P. mexicana*, the maternal ancestor, ranges from the Rio San Fernando southeast of Monterrey, Mexico, to Honduras (*Schlupp, Parzefall & Schartl, 2001*), leading to a mosaic of allopatry and sympatry of the host species with Amazon Mollies. Other members of the genus *Poecilia* also live in Central America and Mexico, but most do not overlap with the range of *P. formosa*. Livebearing fishes are also widespread invasive species; an example is *P. vivipara*, which has populations in South America and introduced populations in the West Indies (*Reznick et al., 2017*). The guppy, *P. reticulata*, has spread across 69 countries on six continents although many of the populations were intentionally introduced for mosquito control (*Sasanami et al., 2021*). The closest related genus to *Poecilia*, *Limia*, does not overlap geographically with *P. formosa* and exists exclusively in the West Indies with several species in Hispaniola, Cuba, Grand Cayman, and Jamaica (*Reznick et al., 2017*; *Rodríguez-Silva, Torres-Pineda & Josaphat, 2020*). The next closest related genera, *Micropoecilia*, *Acanthophacelus*, and *Cnesterodon*, consist of South American species (*Reznick et al., 2017*; *Ramos-Fregonezi, Malabarba & Fagundes, 2017*). Among genera that include sympatric species to *P. formosa*, the most closely related are *Gambusia*, *Heterophallus*, *Belonesox*, *Xiphophorus*, *Heterandria*, *Priapella*. *Carlhubbsia*, *Scolichthys*, *Poeciliopsis*, *Neoheterandria*, *Brachyrhaphis*, *Phallichthys*, *Priapichthys*, *Alfaro*, and *Xenophallus* (*Reznick et al., 2017*). *Girardinus* and *Quintana* species are also related but exist only in Cuba (*Reznick et al., 2017*).

This study tested both whether paternal effects cause differences in reproductive success of *P. formosa* and tested related species to determine the specificity of sperm donors to *P. formosa*. Exploring these questions in *P. formosa* could have broad applicability to paternal non-genetic effects in gynogenetic species and sexually reproducing species as well as reproductive barriers in other hybrid gynogenetic species. Paternal non-genetic effects are difficult to separate from genetic effects in sexually reproducing species, and

sperm-egg interactions during fertilization are poorly characterized (*Okabe, 2018*; *Zigo et al., 2020*; *Vogt, 2021*). In human infertility, available diagnostic tests for sperm morphology and motility do not accurately reflect the fertilization capabilities of sperm, and tests for sperm DNA fragmentation do not result in improved live birth rate (*Kumar & Singh, 2015*; *Cheung et al., 2019*; *Baldini et al., 2021*). Identifying paternal non-genetic effects that affect reproductive outcomes such as pregnancy and live birth rate could lead to better diagnostic tests and treatments for male infertility.

In this study we asked two different questions: first we tested whether reproductive success of Amazon Mollies is related to which male provides the sperm on a population level. Concretely, in Experiment 1 we looked at whether sympatric and allopatric males differed in how many offspring Amazon Molly females had with them. We predicted that the sympatric populations would have greater reproductive success than allopatric populations potentially reflecting the outcome of a coevolutionary process. In another experiment (Experiment 2), we tested natural mating of males from eleven species that are from the family Poeciliidae but were not previously reported to produce offspring successfully with *P. formosa*. We predicted that with phylogenetic distance to the genus *Poecilia* the probability of successful production of offspring with *P. formosa* would diminish.

## MATERIALS AND METHODS

### Ethics statement

All fish care and protocol procedures were performed according to United States federal regulations and institutional policies and approved by the University of Oklahoma Norman Institutional Animal Care and Use Committee (Protocol R21-027). Fish were acquired from the International Stock Center for Livebearing Fishes, University of Oklahoma. Surviving fish were returned to the International Stock Center for Livebearing Fishes, University of Oklahoma. Sample sizes were based on fish availability. Humane endpoints due to spontaneous disease were weight loss or decrease in activity level or appetite, and animals meeting humane endpoints were euthanized using the method described below prior to planned experimental endpoints.

### Environment and timeline

All fish were fed either flake food (TetraMin Tropical Flakes, Spectrum Brands Pet LLC, Blacksburg, VA, USA) or frozen live feed (Hikari Bio-Pure Blood Worms for adults and Hikari Bio-Pure Brine Shrimp for juveniles, Hikari Sales USA Inc., Hayward, CA, USA) once daily for 4 to 6 days per week. Water conductivity was measured at water change and adjusted with Instant Ocean Sea Salt (Spectrum Brands Pet LLC, Blacksburg, VA, USA) to be at least 800 µS/cm. Water temperature was maintained at or above 20 °C, with water temperature from a randomly selected tank sampled multiple times per week. Fish were either housed in 10-liter static tanks or 40-liter static tanks each with a sponge filter, polyvinyl chloride tunnel, pebbles, and live plants as enrichment. Ten-liter tanks had 30% water changes performed weekly and 40-liter tanks had 30% water changes performed every other week. All fish were housed in the same room and experienced a 12:12 h light:

dark cycle. Fish health was monitored daily. Experiments were conducted between March and December 2022.

## Breeding design

Experiment 1 utilized 24 *P. formosa* VI/17 nulliparous (virgin) females from sympatry with *P. mexicana*, each measuring greater than 3 cm standard length, four sympatric *P. mexicana* VI/17 males, and four allopatric *P. mexicana* Oxolotan males. Another 12 *P. formosa* Airport Ditch 2021 (AD21) nulliparous females from sympatry with *P. latipinna*, measuring at least 2.5 cm in standard length, were paired with two sympatric *P. latipinna* AD21 males, or two allopatric *P. latipinna* Steve's Ditch males. Females were observed for pregnancy weekly starting at week 3 after pairing. When a female was visibly pregnant as evidenced by a barrel-shape body, she was isolated in a 10-liter tank until she produced offspring.

In Experiment 2, we paired a total of 24 *P. formosa* nulliparous females, measuring at least 2 centimeters in standard length, with males from several species. We always used two males per species as independent replicates. The species used were *Gambusia affinis*, *Gambusia sexradiata*, *Girardinus metallicus*, *Heterandria formosa*, *Limia islai*, *Limia melanogaster*, *Poeciliopsis prolifica*, *Poecilia dominicensis*, *Poecilia sphenops* from allopatry and *Poecilia sphenops* from sympatry, *Xiphophorus hellerii*, and *Xiphophorus variatus*. Each female was paired individually with one approximately size-matched male in a 10-liter tank until she produced offspring or until the experimental endpoint of 12 or 15 weeks.

No female or male was shared between experiments to minimize the effect of a specific individual's reproductive performance. All experiments utilized two males and at least two females from each population to compensate for variation in individual fertility.

## Reproductive data collection

Pregnant females were checked daily for offspring. On the day that offspring were observed, the mother was humanely euthanized by immersion in (250 mg/L) tricaine methanesulfonate (Tricaine-S, Syndel USA, Ferndale, WA, USA) buffered with (500 mg/L) sodium bicarbonate for 30 min past the cessation of opercular movement. The carcass was then preserved in 4% formaldehyde. One day prior to dissection, the carcass was transferred to tap water to minimize personnel exposure to formaldehyde. The carcass was dissected to count embryos and observe oocytes. The stage of the embryos was classified as early-stage if eyes had not developed and classified as late-stage pregnancy if eyes had developed. Compared to the developmental stage classification of poeciliid species standardized by *Haynes (1995)*, early-stage in this study was equivalent to stages 4–6 and late-stage was equivalent to stages 7–11. The standard length of the female was also measured at time of dissection. Number of offspring was estimated visually on the day of birth, and the number of offspring was confirmed 2 weeks after birth when offspring could safely be moved with a net.

If a female had not produced offspring by the experimental endpoint, the female was humanely euthanized and dissected using the methods described above. The experimental

endpoint for Experiment 1 was 9 to 15 weeks based on at least twice the minimum time from pairing to parturition typically observed. The experimental endpoint for Experiment 2 was 12 to 15 weeks because no females had produced offspring by 9 weeks, but some did between 10 and 12 weeks. Fish were size matched between treatment and control groups, but randomization and blinding were not performed.

## Variable definitions

The groups of interest for Experiment 1 were sympatry (sympatric *vs* allopatric) and female population (VI/17 *vs* AD21). Outcomes for statistical comparison were relative reproductive output, relative embryo output, and combined relative reproductive output. Relative reproductive output was the number of live offspring divided by female standard length in millimeters (*Hinz & Schlupp, 2011*). Relative embryo output was the number of embryos recovered on dissection divided by female standard length in millimeters. Combined relative reproductive output was the sum of relative reproductive output and relative embryo output.

## Statistical analysis

Statistical software used was Microsoft Excel (2016, Microsoft Corporation, Redmond, WA, USA). All statistical tests were two-tailed Mann-Whitney U (MWU) tests, which were performed for each paired combination of variable of interest and outcome in Experiment 1. Statistical significance was set at $\alpha = 0.05$.

## RESULTS

### Experiment 1

For logistical reasons we performed Experiment 1 as two runs that were analyzed separately due to unforeseen differences in fish health. In Experiment 1A, there was no significant difference in female size between the females used with allopatric and sympatric males (U-test, $U = 18$, $n = 12$, $p = 1.0$). All sympatric females had offspring or were pregnant by 9 weeks after breeding began, but only one allopatric female had offspring and no allopatric females were pregnant. The average brood size for sympatric females was 13.7 offspring/late-stage embryos, and the one allopatric female which produced offspring had a brood size of 13 offspring. There was no significant difference in relative reproductive output (U-test, $U = 30.6$, $n = 12$, $p = 1.96$) or relative embryo output (U-test, $U = 12.75$, $n = 12$, $p = 0.4$) between sympatric and allopatric females. However, there was a significant difference in combined relative reproductive output between sympatric and allopatric females (U-test, $U = 2$, $n = 12$, $p = 0.01$) (Fig. 1).

In Experiment 1A, one of the non-pregnant allopatric females died and two more allopatric non-pregnant females were moribund (meeting humane endpoints for decrease in activity level or appetite) and humanely euthanized 4 days before the experimental endpoint at 9 weeks.

Experiment 1B repeated the groupings of Experiment 1A with different fish from the same populations and added groupings of AD21 *P. formosa* females with sympatric AD21 *P. latipinna* males and allopatric Steve's Ditch *P. latipinna* males. In Experiment 1B, there

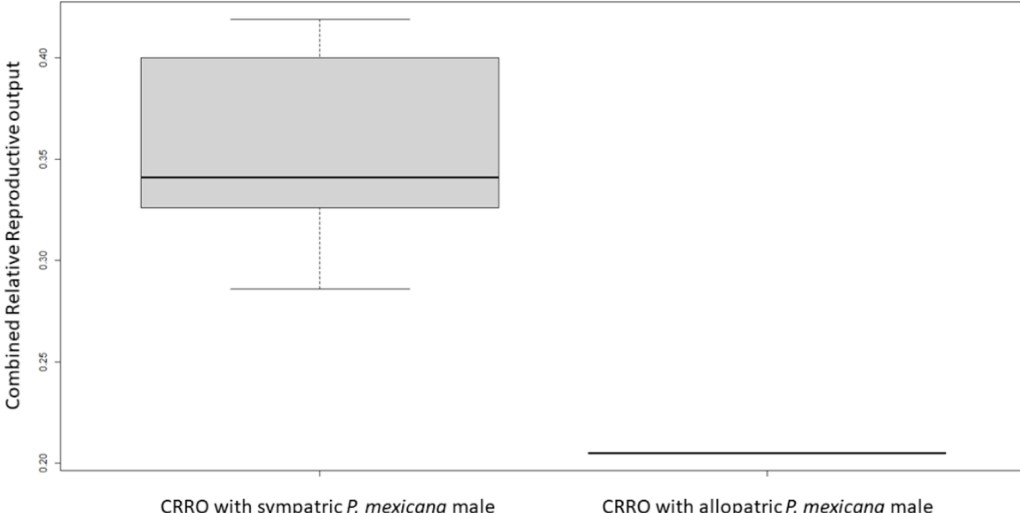

**Figure 1 Boxplot of combined relative reproductive output (CRRO) between sympatric and allopatric VI/17 *Poecilia formosa* in Experiment 1A.** CRRO is equal to the sum of the number of offspring and number of embryos produced divided by the female standard length. CRRO was significantly greater in the sympatric group compared to the allopatric group in Experiment 1A (U-test, $U = 2$, $n = 12$, $p = 0.01$). CRRO did not differ significantly between the sympatric group compared to the allopatric group in Experiment 1B (not shown).

was no significant difference in female size between the sympatric VI/17 females and allopatric VI/17 females (U-test, $U = 14$, $n = 12$, $p = 0.522$). Three allopatric females had offspring with an average brood size of five offspring. Two sympatric females had offspring and embryos with an average brood size of 8.5 offspring/late-stage embryos. There was no significant difference in relative reproductive output (U-test, $U = 23.6$, $n = 12$, $p = 1.63$), relative embryo output (U-test, $U = 39.5$, $n = 12$, $p = 2.0$), or combined relative reproductive output (U-test, $U = 25.1$, $n = 12$, $p = 1.74$). All VI/17 females in Experiment 1B remained healthy for the duration of the experiment. One allopatric male died in the week 2 of the 15-week experiment and was replaced.

Comparing all VI/17 females between Experiments 1A and 1B, there was no significant difference in relative reproductive output (U-test, $U = 119.6$, $n = 24$, $p = 1.99$), relative embryo output (U-test, $U = 125.6$, $n = 24$, $p = 2.0$), or combined relative reproductive output in a two-sided MWU (U-test, $U = 72.6$, $n = 24$, $p = 1.03$). Comparing sympatric VI/17 females between Experiments 1A and 1B, there was no significant difference in relative reproductive output (U-test, $U = 29.5$, $n = 12$, $p = 1.93$) or relative embryo output (U-test, $U = 13.7$, $n = 12$, $p = 0.493$). In Experiment 1A, sympatric VI/17 females had a significantly greater combined relative reproductive output (U-test, $U = 2$, $n = 12$, $p = 0.01$) than sympatric VI/17 females in Experiment 1B in a two-sided MWU. Comparing allopatric VI/17 females between Experiments 1A and 1B, there was no significant difference in relative reproductive output (U-test, $U = 23.6$, $n = 12$, $p = 1.63$), relative embryo output (U-test, $U = 39.5$, $n = 12$, $p = 2.0$), or combined relative reproductive output (U-test, $U = 23.6$, $n = 12$, $p = 1.63$) in a two-sided MWU. There was a significant difference in VI/17 female size between Experiments 1A and 1B (U-test, $U = 5$, $n = 24$, $p < 0.001$) in a

**Table 1 Reproductive data for VI/17 *Poecilia formosa* females crossed with *Poecilia mexicana* males.**
Variables are female standard length (FSL), number of offspring produced, number of embryos produced, relative reproductive output (RRO), relative embryo output (REO), and combined relative reproductive output (CRRO) for each VI/17 female in Experiments 1A and 1B. RRO is equal to the number of offspring produced divided by FSL. REO is equal to the number of embryos produced divided by FSL. CRRO is equal to the sum of RRO and REO. Data from sympatric females is bolded.

| Sympatric/Allopatric | FSL (mm) | Offspring number | Embryo number | RRO | REO | CRRO |
|---|---|---|---|---|---|---|
| **Sympatric** | **43** | **14** | **0** | **0.325** | **0** | **0.325** |
| | **43** | **0** | **18** | **0** | **0.418** | **0.418** |
| | **42** | **12** | **0** | **0.285** | **0** | **0.285** |
| | **41** | **0** | **14** | **0** | **0.341** | **0.341** |
| | **40** | **0** | **16** | **0** | **0.4** | **0.4** |
| | **39** | **0** | **8** | **0** | **0.205** | **0.205** |
| | **34** | **1** | **8** | **0.041** | **0.235** | **0.276** |
| | **33** | **8** | **0** | **0.242** | **0** | **0.242** |
| | **33** | **0** | **0** | **0** | **0** | **0** |
| | **33** | **0** | **0** | **0** | **0** | **0** |
| | **33** | **0** | **0** | **0** | **0** | **0** |
| | **32** | **0** | **0** | **0** | **0** | **0** |
| Allopatric | 45 | 0 | 0 | 0 | 0 | 0 |
| | 43 | 13 | 0 | 0.302 | 0 | 0.302 |
| | 43 | 0 | 0 | 0 | 0 | 0 |
| | 41 | 0 | 0 | 0 | 0 | 0 |
| | 40 | 0 | 0 | 0 | 0 | 0 |
| | 39 | 0 | 0 | 0 | 0 | 0 |
| | 35 | 3 | 0 | 0.085 | 0 | 0.085 |
| | 34 | 2 | 7 | 0.058 | 0.205 | 0.263 |
| | 34 | 3 | 0 | 0.088 | 0 | 0.088 |
| | 34 | 0 | 0 | 0 | 0 | 0 |
| | 34 | 0 | 0 | 0 | 0 | 0 |
| | 33 | 0 | 0 | 0 | 0 | 0 |

two-sided MWU. The VI/17 females in Experiment 1A had an average female standard length of 41.6 mm, while the VI/17 females in Experiment 1B had an average female standard length of 33.5 mm.

In Experiment 1B for AD21 females, none of the females produced offspring and only one allopatric female had a possible early-stage pregnancy with three large, yellow oocytes. Both groups had frequent mortality of the male fish, which were replaced when a male was noted to be sick or dead. Due to the high mortality rate, there were no statistical comparisons performed on relative reproductive output, relative embryo output, or combined relative reproductive output. There was a significant difference in size between the AD21 females and VI/17 females in Experiment 1B (U-test, U = 29.7, $n = 24$, $p = 0.015$), but there was no significant size difference between the sympatric and allopatric AD21 females (U-test, U = 10.3, $n = 12$, $p = 0.22$). The average standard length of an AD21 female

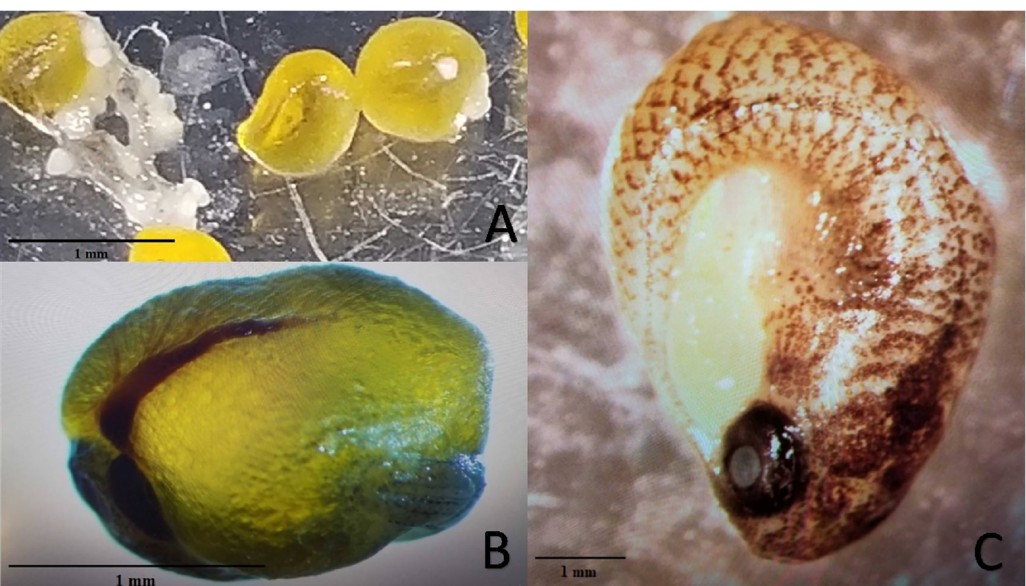

**Figure 2 Examples of mature unfertilized eggs, early-stage embryos, and late-stage embryos.** Photo credit: Clarissa Cerepaka. (A) Mature unfertilized eggs from a pairing of *Poecilia formosa* with *Poeciliopsis prolifica* in Experiment 2. (B) Early-stage embryo from a sympatric cross with *Poecilia formosa* and *Poecilia mexicana* in Experiment 1A. (C) Late-stage embryo from a sympatric cross with *Poecilia formosa* and *Poecilia mexicana* in Experiment 1B. Scale bars indicate estimated size.

in Experiment 1B was 28.58 mm, and the average standard length of a VI/17 female in Experiment 1B was 33.5 mm. Table 1 includes the data for VI/17 females in Experiment 1.

## Experiment 2

Experiment 2 examined reproductive output of pairings between *P. formosa* females and males from species within the genus *Poecilia* and related genera (*i.e. Gambusia, Girardinus, Heterandria, Limia, Poeciliopsis*, and *Xiphophorus*). All pairings with males from the genera *Poecilia* (sympatric and allopatric *P. sphenops, P. dominicensis*) and *Limia* (*L. melanogaster, L. islai*) had at least one pair that produced offspring or late-stage embryos. Two follow-up 15-week pairings between *P. formosa* females and *Poecilia wingei* males from February 2023 to May 2023 in a greenhouse setting did not result in offspring or embryos but did have large, yellow, well-developed oocytes present on dissection of the females. None of the males from other genera (*Gambusia, Girardinus, Heterandria, Poeciliopsis, Xiphophorus*) produced offspring or late-stage embryos. *Poeciliopsis, Girardinus*, and *Xiphophorus* males were seen attempting to mate with the *P. formosa* females. On dissection, one female from each pairing with *P. prolifica, G. metallicus*, and *X. hellerii* appeared to have large, yellow, well-developed oocytes, but the oocytes did not appear to be fertilized because they lacked a visible blastodisc and blood vessels (Fig. 2). In one *G. metallicus* pairing, the male died in the first and fourth week of the 15-week experiment. In one *L. islai* pairing, the male died in week 12 of the 15-week experiment. These two pairs in Experiment 2 that had male mortality did not produce offspring or late-stage embryos despite prompt replacement of the males within a week of the deaths.

**Table 2 Reproductive data for *Poecilia formosa* females crossed with males from *Gambusia, Girardinus, Heterandria, Limia, Poecilia, Poeciliopsis*, and *Xiphophorus* genera.** The species of the male and population of the female in each pairing in Experiment 2, female standard length (FSL), number of offspring produced, number of embryos produced, and combined relative reproductive output (CRRO) are shown. Combined relative reproductive output is equal to the sum of the number of offspring and embryos produced divided by the FSL. Data from species crosses that produced offspring or embryos in at least one pairing are bolded.

| Male species | Female population | FSL (mm) | Offspring number | Embryo number | CRRO |
|---|---|---|---|---|---|
| *Gambusia affinis* | AD21 | 24 | 0 | 0 | 0 |
| *Gambusia affinis* | AD21 | 24 | 0 | 0 | 0 |
| *Gambusia sexradiata* | Wesl | 28 | 0 | 0 | 0 |
| *Gambusia sexradiata* | Wesl | 29 | 0 | 0 | 0 |
| *Girardinus metallicus* | AD21 | 25 | 0 | 0 | 0 |
| *Girardinus metallicus* | AD21 | 26 | 0 | 0 | 0 |
| *Heterandria formosa* | Wesl | 29 | 0 | 0 | 0 |
| *Heterandria formosa* | Wesl | 28 | 0 | 0 | 0 |
| *Poeciliopsis prolifica* | AD21 | 28 | 0 | 0 | 0 |
| *Poeciliopsis prolifica* | AD21 | 31 | 0 | 0 | 0 |
| *Xiphophorus hellerii* | VI/17 | 40 | 0 | 0 | 0 |
| *Xiphophorus hellerii* | AD21 | 32 | 0 | 0 | 0 |
| *Xiphophorus variatus* | AD21 | 24 | 0 | 0 | 0 |
| *Xiphophorus variatus* | AD21 | 27 | 0 | 0 | 0 |
| ***Limia islai*** | **VI/17** | **32** | **5** | **0** | **0.156** |
| ***Limia islai*** | **VI/17** | **27** | **0** | **0** | **0** |
| ***Limia melanogaster*** | **Wesl** | **25** | **1** | **4** | **0.2** |
| ***Limia melanogaster*** | **Wesl** | **25** | **0** | **0** | **0** |
| ***Poecilia dominicensis*** | **Wesl** | **30** | **0** | **6** | **0.2** |
| ***Poecilia dominicensis*** | **AD21** | **24** | **0** | **0** | **0** |
| ***Poecilia sphenops* (allopatric)** | **VI/17** | **32** | **5** | **0** | **0.156** |
| ***Poecilia sphenops* (allopatric)** | **VI/17** | **33** | **0** | **0** | **0** |
| ***Poecilia sphenops* (sympatric)** | **AD21** | **29** | **6** | **0** | **0.206** |
| ***Poecilia sphenops* (sympatric)** | **Wesl** | **20** | **0** | **3** | **0.1** |

A female paired with *P. dominicensis* (combined relative reproductive output = 0.2) and a female paired with *L. melanogaster* (combined relative reproductive output = 0.2) had the greatest combined relative reproductive output (Table 2). Females paired with sympatric *P. sphenops* were the only species cross to have both females have offspring or late-stage embryos. Comparing females who had offspring or embryos from Experiment 2 to females who had offspring or embryos in Experiment 1A, the combined relative reproductive output of females from Experiment 2 was significantly lower (U-test, U = 0, $n$ = 11, $p$ = 0.008) and the female standard lengths were significantly shorter (U-test, U = 2.3, $n$ = 11, $p$ = 0.026). Comparing females who had offspring or embryos from Experiment 2 to females who had offspring or embryos in Experiment 1B, there was no significant difference in combined relative reproductive output (U-test, U = 8, $n$ = 9, $p$ = 0.624), and the female standard lengths did not differ significantly (U-test, U = 2, $n$ = 9, $p$ = 0.05).

## DISCUSSION

### Are there differences in male capability to trigger embryogenesis in Amazon Mollies?

Amazon Mollies are one of a small number of sperm-dependent asexual fishes. They produce clonal offspring without using male genetic contributions. Nonetheless, they require sperm from heterospecific males to trigger embryogenesis. This sperm dependency raises a number of important questions, two of which we addressed in the present study. On the reproductive biological level, we asked if males from allopatric or sympatric populations differ in effectiveness in triggering embryogenesis. We found that there is a non-significant trend for sympatric males to sire more offspring. Based on our design, we cannot distinguish whether this may be a direct effect of the sperm or males, or indirectly a female response. Females may influence the success of fertilization either by behaviorally rejecting mating or through sperm selection by cryptic female choice in the female reproductive tract. When presented with animated *P. mexicana* male models, *P. formosa* preferred models with lateral projection areas similar to those of sympatric males, but it is unknown if the behavioral preference would result in more mating or more offspring (*Kim et al., 2014*). Cryptic female choice has not been studied in *P. formosa*. However, in a recent zebrafish study, female reproductive fluid preferentially attracted spermatozoa with longevity greater than 20 s and better DNA integrity (*Cattelan et al., 2023*). Our study was limited by small sample sizes, due to logistical reasons and unexpected morbidity and mortality. Nonetheless, if correct, our finding of differences between the sympatric and allopatric pairings would point to coevolution of Amazon Mollies and their host males by suggesting that sympatric donor sperm or the receiving Amazon Mollies have adapted to activate embryogenesis more successfully during mating. Something similar has been reported by *Gabor, Ryan & Morizot (2005)*, who found reproductive character displacement in Sailfin Mollies depending on sympatry or allopatry with Amazon Mollies. Based on these findings one would predict *very high specificity* of the sperm-egg interaction in Amazon Mollies.

Our study did not confirm this prediction, but our findings had several limitations. We ran the experiments in two independent rounds, which may have introduced effects of seasonality and the females used differed in size. Based on the data collected, larger *P. formosa* (measuring 40 mm in female standard length) may be more suitable for detecting subtle differences in reproductive success as they appear more likely to successfully produce offspring and embryos, even though *P. formosa* as small as 25 mm in female standard length occasionally produced live offspring. Further studies of the relationship between female standard length and reproductive output could develop new coefficients or equations to refine comparisons of reproductive output between *P. formosa* of different sizes. Although seasonality of reproduction has not been studied thoroughly in *P. formosa*, *Robinson et al. (2011)* studied the mating season of a parental species *P. latipinna* in Texas. The percentages of gravid *P. latipinna* females were more than 4.5 times greater in the early mating season of March and April and the mid mating season of May through August than in the late mating season of September and October. Out of

the 988 female *P. latipinna* sampled, 4.8% had embryos present in November through February, 69.3% had embryos present in March and April, 66.4% had embryos present in May through August, and 14.7% had embryos present in September and October (*Robinson et al., 2011*). To the contrary, the highest gonadosomatic index for females of the other parental species *P. mexicana* occurred between August and February, in a population near Veracruz, Mexico. However, this sample only included 100 *P. mexicana* females and gonadosomatic index was based on many criteria other than presence of embryos, such as weight, standard length, weight of ovary, number of mature oocytes, and number of embryos (*Chávez-López, Rocha-Ramírez & Cortés-Garrido, 2015*). Behavioral interactions may also be important in this context: *Schlupp & Plath (2005)* found that *P. mexicana* males did mate with *P. mexicana* females significantly more than *P. formosa* females over 30 min and that *P. mexicana* females received significantly more sperm. *Makowicz, Muthurajah & Schlupp (2018)* found that neither *P. mexicana* nor *P. latipinna* males showed a behavioral preference between sympatric and allopatric *P. formosa*, and *P. formosa* did not show a behavioral preference between *P. mexicana* and *P. latipinna*. These findings suggest that although *P. mexicana* males can differentially allocate sperm and choose conspecifics, *P. mexicana* males may not breed more effectively with sympatric *P. formosa* females than allopatric *P. formosa* females. If the significant difference in VI/17 *P. formosa* combined relative reproductive output favoring sympatric populations of *P. mexicana* is a true difference, then the difference is quite possibly at the epigenetic level rather than driven by male behavior, based on the findings of *Makowicz, Muthurajah & Schlupp (2018)*.

## Heterospecific matings with males throughout the family Poeciliidae

Considering which other species are able to provide sperm provides a different perspective on sperm specificity. Based on our data and previous studies (summarized in Table 3), sperm-egg interactions are *not very specific*, as a large number of species can trigger embryogenesis in Amazon Mollies. It appears that every species of the genus *Poecilia* that was tested so far is capable of triggering embryogenesis with the possible exception of *Poecilia wingei*. *Poecilia wingei* is closely related to *Poecilia reticulata*, which does successfully produce offspring with Amazon Mollies (*Hubbs & Hubbs, 1946a*; *Schlupp, Parzefall & Schartl, 2001*). The taxonomy of this group is unfortunately not fully resolved, but the major groups are well defined. If one breaks down the genus *Poecilia* into *Acanthocephalus* (the guppy and relatives), *Allopoecilia*, *Mollienesia* (Mollies *sensu stricto*), *Limia*, *Pamphorichthys*, and *Micropoecilia*, only *Pamphorichthys* has not been tested yet. The genus *Limia*, which is the sister group to *Mollienesia*, is also clearly capable of triggering embryogenesis. Specifically, in the present study we tested the species that is at the base of the genus, *L. melanogaster* and obtained offspring, as with all other species of *Limia* that were tested so far. When consulting the phylogeny by *Reznick et al. (2017)*, the genus *Cnestorodon* would be a good next choice to further investigate the role of phylogenetic distance in sperm specificity in the Amazon Molly. While many species of livebearing fishes can successfully provide sperm, this is not true for all species or genera in the family (Fig. 3). Our study and previous ones also considered species that show

**Table 3 Known experimental sperm donors to Amazon mollies (*Poecilia formosa*) in natural mating crosses.** Compared data from Experiment 2 to two previous publications with the most comprehensive lists of natural mating crosses with *P. formosa* (*Hubbs & Hubbs, 1946a*; *Schlupp, Parzefall & Schartl, 2001*). Information on *Limia vittata* reproduction is from *Schlupp et al. (1996)*. "Yes" indicated that live offspring were produced with the exception of *Poecilia dominicensis*, which only produced late-stage embryos. "No" indicated that no live offspring resulted from the crosses. Crosses varied in duration of pairing from 84 days to 1,628 days. Names from *Hubbs & Hubbs (1946a)* are given in parentheses next to their respective currently accepted species names (*Huber, 2022*, Killi-Data online; *Kallman & Kazianis, 2006*). Species or genera crosses that did produce offspring in at least one study are bolded. There are conflicting findings between the present study and *Hubbs & Hubbs (1946a)*, which may partially be explained by the experimental manipulations of the Gambusia and *P. formosa* cross alluded to in *Hubbs & Hubbs (1946b)*.

| | Hubbs-1946 | Schlupp-1996 | Schlupp-2001 | Present study | All studies |
|---|---|---|---|---|---|
| ***Poecilia sphenops (Molliensia sphenops)*** | Yes | | | Yes | Yes |
| ***Poecilia butleri (Molliensia acapulco)*** | Yes | | | | Yes |
| ***Poecilia latipunctata*** | | | Yes | | Yes |
| ***Poecilia velifera (Molliensia velifera)*** | Yes | | Yes | | Yes |
| ***Poecilia petenensis*** | Yes | | Yes | | Yes |
| ***Poecilia catemaconis*** | | | Yes | | Yes |
| ***Poecilia gilli*** | | | Yes | | Yes |
| ***Poecilia sulphuraria*** | | | Yes | | Yes |
| ***Poecilia reticulata ( Lebistes* species)** | Yes | | Yes | | Yes |
| ***Poecilia dominicensis*** | | | | Yes | Yes |
| ***Poecilia caucana ( Allopoecilia* species)** | Yes | | | | Yes |
| ***Limia islai*** | | | | Yes | Yes |
| ***Limia melanogaster*** | | | | Yes | Yes |
| ***Limia vittata*** | | Yes | | | Yes |
| ***Gambusia* species** | Yes | | | | Yes |
| ***Gambusia affinis*** | Yes? | | | No | No |
| *Gambusia sexradiata* | | | | No | No |
| *Quintana atrizona (Quintana* species) | No | | | | No |
| *Xiphophorus* species (*Platypoecilus* species) | No | | No | No | No |
| *Xiphophorus hellerii* | | | No | No | No |
| *Xiphophorus variatus* | | | No | No | No |
| *Heterandria* species | No | | | No | No |
| *Heterandria formosa* | | | | No | No |
| *Girardinus* species (*Glaridichthys* species) | No | | | No | No |
| *Girardinus metallicus* | | | | No | No |
| *Poeciliopsis* species (*Poecilistes* species) | No | | | No | No |
| *Poeciliopsis prolifica* | | | | No | No |
| *Micropoecilia* species | No | | | | No |

geographical overlap with Amazon Mollies that are only distantly related. This includes for example the speciose genera *Xiphophorus* and *Poeciliopsis*, both of which seem unable to trigger embryogenesis in Amazon Mollies. Our work specifically tested a representative of the genus *Girardinus*, which is basal to the clades that are unable to trigger embryogenesis. A notable exception is reported in the article by *Hubbs & Hubbs (1946a)*, where they claim to have found reproduction with an unspecified species of *Gambusia*. In a meeting

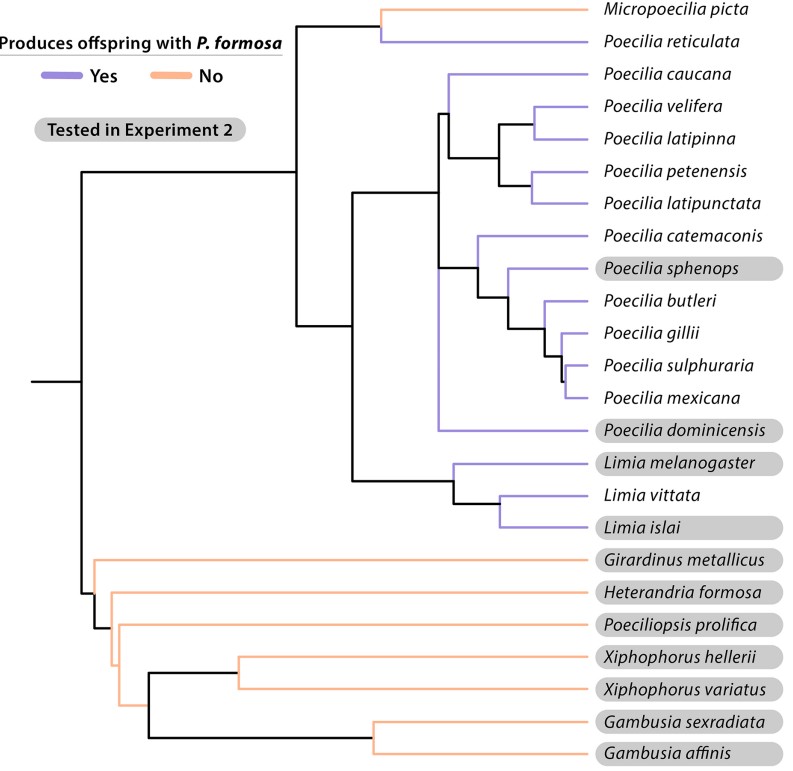

**Figure 3 Cladogram of taxonomic relationships of species experimentally crossed with the Amazon Molly (*Poecilia formosa*).** Most relationships given were based on topology from *Chang et al. (2019)* and phylogeny developed by *Furness et al. (2021)*. Additional *P. dominicensis*, *P. velifera*, and *P. catemaconis* relationships were adapted from *Palacios et al. (2016)*. Additional *L. islai* relationships were adapted from *Spikes et al. (2021)*. Information on *L. vittata* reproduction is from *Schlupp et al. (1996)*. Species that successfully produce late-stage embryos or live offspring when crossed with *P. formosa* are listed in blue. Species that fail to reproduce with *P. formosa* are listed in red. Species tested in Experiment 2 are highlighted in gray. Additional reproductive data is from Table 3.

abstract, *Hubbs & Hubbs (1946b)* clarified that offspring from crosses with *Gambusia* were principally obtained through use of pituitary injections and artificial insemination. Our study, which examined natural mating using *G. affinis* and *G. sexradiata*, was unable to confirm the previous finding, but additional work on this would be useful. Overall, we confirmed that *P. formosa* is not compatible in natural mating with *X. hellerii* or *X. variatus*. Additional new species tested belonged to the genera *Gambusia, Girardinus, Heterandria*, and *Poeciliopsis*, but none of the pairings produced live offspring or late-stage embryos.

It should also be noted that the taxonomy of the mollies is in flux and that a recent phylogeny by *Palacios et al. (2023)* classified the sympatric *Poecilia* species from Baños de San Ignacio as *P. mexicana* rather than *P. sphenops*. A similar problem is associated with the article by *Hubbs & Hubbs (1946a)*, where we carefully tried to align old taxonomy with the currently accepted. *Gambusia* species were previously reported by *Hubbs & Hubbs (1946a, 1946b)* to produce live offspring with *P. formosa*, likely with *G. affinis* using artificial insemination and pituitary hormone treatment of the female. In the same

abstract, *Hubbs & Hubbs (1946b)* claim "fecundity was roughly proportional to degree of relatedness". It is not clear what direction that relationship has had (presumably more offspring with closer related taxa), but we were not able to corroborate this finding. Nonetheless, the notion that artificial insemination may reveal even more species as potential sperm donors, is of critical importance as it points to prezygotic isolating mechanisms to be potentially behind the currently known pattern. Prezygotic factors could include behavioral or physical barriers to intercourse, incompatibility of the gonopodium with the gonopore of the female, spermatozoa unable to travel to the oocyte, failure of fertilization due to lack of capacitation of the spermatozoon or binding to the vitelline membrane, or failure of embryogenesis by inappropriate epigenetic or other biochemical signaling for successful cell divisions. Based on *Hubbs & Hubbs (1946b)* and our study, an experiment that would test the role of pre- and postzygotic isolation using artificial insemination would be useful.

### Connection to infertility and future research

Our experiments confirm that fertilization in *P. formosa* is fairly non-specific as species from the genera of *Poecilia* and *Limia*, including multiple allopatric species and species that diverged more than two million years ago (*Palacios et al., 2016*), can produce offspring with *P. formosa* through mating in a laboratory setting. If techniques such as artificial insemination could successfully bypass prezygotic barriers, fertilization may prove to be even less specific. With removal of the outer *zona pellucida* of the oocyte, sperm-egg fusion between species as disparate as humans and Syrian hamsters (*Mesocrecitus auratus*) supports that sperm-egg interactions during fertilization may not be highly species specific (*Yanagimachi, Yanagimachi & Rogers, 1976*; *Berros et al., 1978*; *Aitken, 2006*). Unlike in most sexually reproducing species, sperm-egg fusions in *P. formosa* with sperm from other species can result in embryo development and live viable offspring. Further study of the differences between sperm that result in live offspring compared to sperm that does not result in embryo development can help identify non-genetic components of sperm, such as messenger RNA, long non-coding RNA, or short non-coding RNA, associated with embryo development. Some sperm RNAs are differentially expressed between fertile and infertile men, but it is difficult to extrapolate association to causation in humans (*Santiago et al., 2022*). If important sperm RNAs are identified using a *P. formosa* model, the identified RNAs would be a good target for experiments microinjecting identified RNAs into a mouse zygote to confirm epigenetically driven differences in fertility and offspring phenotype (*Grandjean et al., 2015*). Because many RNAs are highly evolutionarily conserved between fish and mammals (*Ord et al., 2020*), *P. formosa* may be a valuable model for translational research to human infertility.

## ACKNOWLEDGEMENTS

We are grateful to many members of the Schlupp lab for help with fish care, especially Ben Conard and Alex Grimsley and to Tyler Reich and Waldir Miron Berbel de Filho for comments on the manuscript. We thank Dunja Lamatsch for helpful suggestions on the

manuscript and Cameron Siler for assistance with the cladogram in Fig. 3. This article is in partial fulfillment of the requirements for a Master's degree for C. C.

### Funding
Financial support for publication was provided by the University of Oklahoma Libraries. The funders had no role in study design, data collection and analysis, decision to publish, or preparation of the manuscript.

### Grant Disclosures
The following grant information was disclosed by the authors:
University of Oklahoma Libraries.

### Competing Interests
The authors declare that they have no competing interests.

### Author Contributions
- Clarissa Cerepaka conceived and designed the experiments, performed the experiments, analyzed the data, prepared figures and/or tables, authored or reviewed drafts of the article, and approved the final draft.
- Ingo Schlupp conceived and designed the experiments, prepared figures and/or tables, authored or reviewed drafts of the article, and approved the final draft.

### Animal Ethics
The following information was supplied relating to ethical approvals (*i.e.*, approving body and any reference numbers):

The University of Oklahoma Institutional Animal Care and Use Committee provided full approval for this research.

### Data Availability
The raw measurements from each Amazon Molly for offspring numbers, embryo numbers, oocyte numbers, and female standard length are available in the Supplemental File.

### Supplemental Information
Supplemental information for this article can be found online at http://dx.doi.org/10.7717/peerj.16118#supplemental-information.

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
