# Peer review of "Sperm specificity and potential paternal effects in gynogenesis in the Amazon Molly (Poecilia formosa)"

_PeerJ, doi:10.7717/peerj.16118_

## Round 0.1 · original submission · Minor Revisions

Thank you for submitting your manuscript to PeerJ. The two reviewers of your paper were positive about its design, interpretation and conclusions. They suggested only a few areas for consideration.

Reviewer 1 lists some suggested editing and asked if you could add scale bars to your images in figure 2 if possible (note that Reviewer 1 attached their review as a PDF annotation). Reviewer 2 raises a point about female mate choice that could be addressed in your discussion. I have one question for you. You state that data from your experiments are summarized in tables 1 and 2. If these tables include all data needed for a reader to do a statistical analysis on your results, I would suggest stating that these tables “include” the data from the experiments. If instead some data are excluded, could these be included in a supplemental table?

I invite you to submit a revised manuscript after minor revisions and ask that your rebuttal letter address all reviewer comments.

I look forward to receiving your revised submission.

·

Basic reporting

No comment.

Experimental design

No comment.

Validity of the findings

No comment.

Reviewer 2 ·

Basic reporting

This is a well written, and easy to follow article. The authors clearly outline their research questions with sufficient motivation. I altogether found everything to be in order in regard to the references, figures and presentation of the manuscript.

Experimental design

The authors lay out a clear motivation for wanting to test this question. They're interested in which species the gynogenetic Amazon molly will/can use to stimulate embryogenesis. The main question is to understand how genetic relatedness (more closely related species versus less closely related) in addition to allopatric versus sympatric species influences the Amazons' ability to reproduce with certain males. The experimental design is straightforward and seems robust. It's a simple design that is clearly described - pairing up Amazon females with males of different populations/species.

Validity of the findings

The results seem straightforward and very clear - Amazons are more likely to produce offspring with sympatric males and males from more closely related genera. Conclusions are well stated and do not overreach the results themselves.

Additional comments

Altogether this was a straightforward and self-contained study. The questions are clear and the results are clear. The only potential modification I could see would potentially be a discussion of how much female choice/actions plays into this. The authors paired up the females with a given male to allow him access to her, but this makes the assumption that the males are equally able and the females equally likely to allow insemination by the males. It's my understanding that the females are typically quite larger than the males and can often fend him off if she doesn't prefer him. So it may not be that the sperm from particular males is physically incapable of stimulating embryogenesis but rather that females prevent some males from inseminating her altogether. This could just be something that gets a bit more attention in the discussion. Otherwise, it's a nice contribution that helps build up our general knowledge of reproduction in this species.

---

## Round 0.2 · accepted · Accept

Thank you for your consideration of and response to all reviewer comments. Based on your revision I am happy to now accept your paper for publication in PeerJ.

Thank you for choosing PeerJ as a venue for publishing your work.

·

Basic reporting

No comment.

Experimental design

No comment.

Validity of the findings

No comment.

Additional comments

No comment.